# Selective Sensing: A Data-driven Nonuniform Subsampling Approach for Computation-free On-Sensor Data Dimensionality Reduction

## Abstract

Designing an on-sensor data dimensionality reduction scheme for efficient signal sensing has always been a challenging task. Compressive sensing is a state-of-the-art sensing technique used for on-sensor data dimensionality reduction. However, the undesired computational complexity involved in the sensing stage of compressive sensing limits its practical application in resource-constrained sensor devices or high-data-rate sensor devices dealing with high-dimensional signals. In this paper, we propose a selective sensing framework that adopts the novel concept of data-driven nonuniform subsampling to reduce the dimensionality of acquired signals while retaining the information of interest in a computation-free fashion. Selective sensing adopts a co-optimization methodology to co-train a selective sensing operator with a subsequent information decoding neural network. We take image as the sensing modality and reconstruction as the information decoding task to demonstrate the 1st proof-of-concept of selective sensing. The experiment results on CIFAR10, Set5 and Set14 datasets show that selective sensing can achieve an average reconstruction accuracy improvement in terms of PSNR/SSIM by 3.73dB/0.07 and 9.43dB/0.16 over compressive sensing and uniform subsampling counterparts across the dimensionality reduction ratios of 4-32x, respectively. Source code is available at `https://figshare.com/s/519a923fae8f386d7f5b`.

## 1 Introduction

In the era of Internet-of-things (IoT) data explosion (Biookaghazadeh et al., 2018), efficient information acquisition and on-sensor data dimensionality reduction techniques are in great need. Compressive sensing is the state-of-the-art signal sensing technique that is applicable to on-sensor data dimensionality reduction. However, directly performing compressive sensing in the digital domain as a linear transformation of signals can be computationally costly, especially when the signal dimension $n$ is high and/or a data-driven sensing matrixMousavi et al. (2017; 2018); Lohit et al. (2018); Wu et al. (2018) is used. To mitigate this problem, several approaches have been proposed to reduce the computational complexity of compressive sensing by constraining the sensing matrices to be sparse, binary, or ternary (Wang et al., 2016; Nguyen et al., 2017; Zhao et al., 2018; Hong et al., 2019). While these approaches can reduce the computational complexity by a constant factor ($O(cn^2)$, where $c$ can be as low as $10^{-2}$), such reduced computational complexity can be still too high to be affordable for resource-constrained sensor devices, *e.g.*, low-cost IoT sensors (Djelouat et al., 2018), or high-data-rate sensor devices dealing with high-dimensional signals, *e.g.*, LiDAR and depth map (Chodosh et al., 2019). Other approaches (Duarte et al., 2008; Robucci et al., 2010) propose to implement compressive sensing in the analog domain instead, eliminating or reducing the computation cost of compressive sensing through custom hardware implementation. However, such custom hardware implementation inevitably increases the cost of the sensor and is often specific to the sensor design, thereby cannot be generally applied to other sensors or applications.

In this paper, we propose a selective sensing framework to address the above-mentioned problem by adopting the novel concept of data-driven nonuniform subsampling to reduce the dimensionality of acquired signals while retaining the information of interest in a computation-free fashion. Specifically, the data dimensionality reduction in selective sensing is a nonuniform subsampling (or

selection) process that simply selects the most informative entries of a signal vector based on an optimized, stationary selection index vector informed by training data. Since no computation is involved for any form of data encoding, the computational complexity of the selective sensing operator is simply $O(1)$, leading to the computation-free data dimensionality reduction during the selective sensing process.[1] Selective sensing adopts a co-optimization methodology to co-train a selective sensing operator with a subsequent information decoding neural network. As the trainable parameters of the sensing operator (the selection index) and the information decoding neural network are discrete- and continuous-valued, respectively, the co-optimization problem in selective sensing is a mixed discrete-continuous optimization problem that is inherently difficult to solve. We propose a feasible solution to solve it by transforming the mixed discrete-continuous optimization problem into two continuous optimization subproblems through interpolation and domain extension techniques. Both of the subproblems can then be efficiently solved using gradient-descent-based algorithms. We take images as the sensing modality and reconstruction as the information decoding task to demonstrate the 1st proof-of-concept of selective sensing. The experiments on CIFAR10, Set5 and Set14 datasets show that the selective sensing framework can achieve an average reconstruction accuracy improvement in terms of PSNR/SSIM by 3.73dB/0.07 and 9.43dB/0.16 over compressive sensing and uniform subsampling counterparts across the dimensionality reduction ratios of 4-32x, respectively.

The contributions of this paper are summarized as follows:

1. We propose a new on-sensor data dimensionality reduction method called selective sensing. Selective sensing efficiently reduces the dimensionality of acquired signals in a computation-free fashion while retaining information of interest. The computation-free nature of selective sensing makes it a highly suitable solution for performing on-sensor data dimensionality reduction on resource-constrained sensor devices or high-data-rate sensor devices dealing with high-dimensional signals.

2. We propose and apply the novel concept of data-driven nonuniform subsampling. Specifically, we first formulate the problem of co-optimizing a selective sensing operator with a subsequent information decoding neural network as a mixed discrete-continuous optimization problem. Furthermore, we propose a viable solution that transforms the problem into two continuous optimization subproblems that can be efficiently solved by gradient-descent-based algorithms, which makes the co-training feasible.

3. We empirically show that data-driven nonuniform subsampling can well preserve signal information under the presence of a co-trained information decoding network.

## 2  RELATED WORK

### 2.1  NONUNIFORM SUBSAMPLING

Model-based nonuniform subsampling has been proposed in Chepuri et al. (2016) in the name of sparse sensing. Sparse sensing requires a hand-crafted sparsity model of a signal as prior knowledge. Differently, selective sensing requires no prior knowledge about the sparsity model of a signal, as all the necessary information needed for reconstruction can be learned from data through the training process. Therefore, selective sensing has a much broader range of applications, especially in IoT, than sparse sensing, considering a vast majority of IoT signals are not well studied nor understood yet, but huge amounts of IoT data are already available for training and learning. Dadkhahi & Duarte (2014) proposes to generate an image mask that can preserve the manifold structure presented in image data. Differently, we focus on the task of single image sensing and reconstruction in this paper. Baldassarre et al. (2016); Weiss et al. (2019); Gözcü et al. (2018); Bahadir et al. (2019; 2020) propose to perform MRI image nonuniform subsampling in k-space (frequency domain). As many spatial-domain signals are much sparser in the frequency domain, *e.g.*, natural images and MRI images, the existing nonuniform subsampling approaches performed in k-space are insufficient

---

[1]For temporal signals, the selection operation can be simply implemented in the digital domain with a counter and a mux that already exists in the control logic of most sensors. We consider such operations as control rather than data computation as no data is computed during the selective sensing process. For spatial signals such as images, the selective sensing operator can also be implemented as a low-cost masked sensor array with no computation involved. In addition, Mayberry et al. (2014); Centeye (2020) present image sensor architectures for embedded systems that can provide pixel-level control of image sensors.

for dealing with dense signals directly in the spatial domain. In addition, the complex computation of or the custom hardware(Macfaden et al., 2017) for implementing Fourier transformation required in these methods is a deal-breaker for resource-constrained sensor devices and/or high-data-rate sensor devices dealing with high-dimensional signals. Differently, selective sensing works directly in the spatial domain and the selective sensing operators require no computation upon the sensor data at all. Huijben et al. (2019) propose to co-optimize a probabilistic subsampling mask and a subsequent task-specific neural network in an end-to-end fashion. The sensing mask is dynamically generated from a random distribution with respect to each signal. The computation of generating such masks is hardly affordable for resource-constrained and/or high-data-rate sensor devices. Differently, selective sensing uses a static sensing mask learnt through the co-training algorithm. Once a sensing mask is depolyed to sensor devices, no computation is needed to update the existing mask. Therefore, selective sensing is extremely friendly to resource-constrained or high-data-rate sensors.

## 2.2 SENSING MATRIX SIMPLIFICATION METHODS

The computational complexity of the linear transformation in compressive sensing is $O(n^2)$. Zhao et al. (2018); Hong et al. (2019) proposes model-based methods to construct sparse sensing matrices. Wang et al. (2016); Nguyen et al. (2017) propose data-driven methods to build binary or ternary sensing matrices. However, all these approaches could only reduce the computational complexity by constant factors, i.e. $O(cn^2)$, where $c$ can be as low as $10^{-2}$). A key differentiator of selective sensing is that by adopting the novel concept of data-driven non-uniform subsampling, it is computation-free and has a computational complexity of $O(1)$.

## 2.3 DATA-DRIVEN COMPRESSIVE SENSING

Kulkarni et al. (2016); Mousavi & Baraniuk (2017); Yao et al. (2019) propose to directly learn the inverse mapping of compressive sensing through the training of reconstruction neural network models. In addition, Mousavi et al. (2017; 2018); Lohit et al. (2018); Wu et al. (2018) propose to co-train a customized sensing scheme with a reconstruction neural network to improve the reconstruction accuracy. It should be noted that such co-training algorithms are specific to the reconstruction network proposed in the corresponding literatures. To the best of our knowledge, there is no general co-training algorithm that can be applied to various reconstruction networks in the domain of data-driven compressive sensing. These approaches inspire us to develop a framework that co-trains a selective sensing operator and a subsequent information decoding network. Co-trained signal sensing and reconstruction frameworks can be viewed as a specific type of autoencoders(Goodfellow et al., 2016). The main difference between such frameworks and a general autoencoder model is that the sensing (encoder) part of such frameworks must be implemented on sensors for on-sensor data dimensionality reduction. Therefore, the computation complexity of the encoder has to be extremely low in order to be affordable for sensor devices.

## 2.4 IMAGE SUPER-RESOLUTION

The problem of neural-network-based image super-resolution has been studied in recent years(Ledig et al., 2017; Dong et al., 2015; Yang et al., 2019). The image super-resolution task is fundamentally different from the image reconstruction task of selective sensing in following two aspects. First, images in super-resolution tasks are uniformly subsampled in the training phase, while images in selective sensing are nonuniformly subsampled. Therefore, the existing network structures for image super-resolution cannot be directly applied to perform the image reconstruction task in selective sensing. Second, the downsizing factor of images in super-resolution tasks is only up to 4x to the best of our knowledge in the existing literature. Differently, in selective sensing and reconstruction tasks, the nonuniformly subsampling factor (compression ratio) of images can have a much larger range (4-32x in this paper).

## 3 METHODOLOGY

In this section, we first formulate the co-optimization of a selective sensing operator and a subsequent information decoding network as a mixed discrete-continuous optimization problem. Then, by applying continuous interpolation and domain extension on the integer variables, we reformulate the

mixed discrete-continuous optimization problem into two continuous optimization problems, both of which can be solved by conventional gradient-descent-based algorithms. Based on the new formulation, we extend the conventional backpropagation(BP) algorithm to derive a general co-training algorithm to co-optimize a selective sensing operator and a subsequent information decoding network. At last, by taking images as the sensing modality and using reconstruction as the information decoding task, we propose a practical approach, referred to as SS+Net, to compose a selective sensing framework for image selective sensing and reconstruction.

In this paper, a lowercase letter denotes a scalar or a scalar-valued function, and a uppercase letter denotes a vector, a matrix, a tensor, or a vector-valued function. We use brackets to index the element of a vector, a matrix, or a tensor. For example, assume $X$ denotes a $n$-dimensional vector $X = [x_0, ..., x_{n-1}]$, then $X[i] = x_i$ for $i = 0, \cdots, n-1$.

### 3.1 PROBLEM FORMULATION

Consider the original signal $X$ is an $n$-dimensional vector, the subsampling rate is $\frac{m}{n}$, and the subsampled measurement $Y$ is a $m$-dimensional vector. The selective sensing of $X$ is a nonuniform subsampling or a selection process that can be formulated as

$$Y = S(X, I) = [X[I[0]], \cdots, X[I[m-1]]], \tag{1}$$

where $S(X, I)$ is a function that stands for the selective sensing operator. $I$ is a $m$-dimensional vector denoting the selection set, which contains the indices (integer values between 0 and $n-1$) of the elements to be selected. Consider $N(Y, \Theta)$ is a subsequent information decoding network and $\Theta$ is the trainable parameters. $o$ is a differentiable objective function that measures the information loss throughout the entire selective sensing process with respect to a information acquisition task. For instance, in a signal reconstruction task, the objective function can be defined as a loss function which measures the difference between the reconstructed signal and the original signal. The co-optimization problem of the sensing operator $S$ and the information decoding network $N$ can be formulated as

$$I_{opt}, \Theta_{opt} = \arg\min_{I, \Theta} o(N(S(X, I), \Theta)),$$
$$\text{subject to } i_0, \ldots, i_{m-1} \text{ are integers within interval } [0, n-1] \tag{2}$$

Given the entries of $\Theta$ are continuous variables, and the entries of $I$ are constrained to be integer variables within $[0, n-1]$, the problem in (2) is a mixed discrete-continuous optimization problem that can not be directly solved with conventional gradient-descent-based algorithms. This is because the gradient of $o$ with respect to $I$ does not exist.

### 3.2 REFORMULATION BY CONTINUOUS INTERPOLATION AND DOMAIN EXTENSION

By applying the continuous interpolation on $S$ with respect to $I$ and the extension on the domain of $S$, we can reformulate the problem in (2) into two subproblems. For simplicity, we adopt a linear interpolation in our method. However, nonlinear interpolation methods can be also applied.

We define a piece-wise linear function $f(X, i)$ as

$$f(X, i) = (X[r_u] - X[r_d])(i - i_d) + X[r_d],$$
$$where\ i_u = floor(i) + 1,\ i_d = floor(i),\ r_u = i_u \bmod n \text{ and } r_d = i_d \bmod n. \tag{3}$$

In (3), $i$ is a real-valued scalar, $floor()$ is the flooring function returning the closest integer that is less than or equal to the input, and $\bmod$ is the modulo operation. $f(X, i)$ essentially interpolates the value of $X$ over a continuous index $i$ in a piece-wise linear fashion and extends the range of $i$ to $(-\infty, \infty)$. Given a $X$, $f(X, i)$ is periodic over every $n$-length interval of $i$. At integer values of $i$, we have $f(X, i) = X[i \bmod n]$, which returns the original value of the $[i \bmod n]$-th element of $X$. Specifically, when $i$ is an integer in interval $[0, n-1]$, we have $f(X, i) = X[i]$. Due to the continuous interpolation and domain extension, $f(X, i)$ is almost everywhere differentiable over $i$ except for all the integer points. The choice of the gradient value at integer points turn out to be insensitive to the algorithm performance. For simplicity, we define the derivatives of $f(X, i)$ at integer values of $i$ as zero. As such, we have the gradient value of $f$ with respect to $i$ in the whole space which can be expressed as

$$\frac{\partial f}{\partial i} = \begin{cases} 0, & \text{if } i \text{ is a integer,} \\ (X[r_u] - X[r_d]), & \text{otherwise.} \end{cases} \tag{4}$$

Based on (4), we define a continuous selective sensing operator function $S'$ as

$$S'(X, I) = [f(X, i_0), ..., f(X, i_{m-1})]. \tag{5}$$

Leveraging (3) and (5), we reformulate the mixed discrete-continuous optimization problem in (2) into two subproblems defined as

$$I_R, \Theta_R = \arg\min_{I, \Theta} o(N(S'(X, I), \Theta)), \tag{6}$$

and

$$I_{opt} = [round(i) \bmod n \text{ for each entry } i \text{ in } I_R], \\ \Theta_{opt} = \arg\min_{\Theta} o(N(S(X, I_{opt}), \Theta)), \tag{7}$$

where $round()$ is an even rounding function that returns the closest integer value of the input, and the initial values of $\Theta$ in (7) is $\Theta_R$. Note that both the subproblems in (6) and (7) are unconstrained and the gradient of $o$ with respect to $I$ and $\Theta$ can be calculated over the whole space in (6). Therefore, we can solve the subproblems in (6) and (7) sequentially using gradient-descent based algorithms. For the brevity of illustration, we refer to the process of solving the subproblem in (6) and (7) as the initial-training and the fine-tuning step, respectively, in the rest of this paper.

### 3.3 EXTENSION OF THE BACKPROPAGATION ALGORITHM

Generally, neural network models are trained over multiple training samples and the gradients of trainable parameters are calculated using the BP algorithm. We extend the BP algorithm and derive the gradient calculation (with respect to $I$) over a batch of training samples as follows.

Given a batch of $b$ samples $X_1, \cdots, X_b$ of the signal $X$ for training, the forward pass of the BP algorithm in the initial-training step can be derived as

$$Y_i = S'(X_i, I), \ Z_i = N(Y_i, \Theta), \ o_i = o(Z_i), \ o_{batch} = \frac{1}{b} \sum_{i=1}^{b} o_i, \tag{8}$$

where $i = 1, \cdots, b$, $Z_i$ is the representation of the information decoded by the network and $o_{batch}$ is the loss function that measures the average information loss throughout the selective sensing process. One can also choose to use the total information loss here.

In the backward pass of the BP algorithm, the gradient calculation with respect to $\Theta$ is the same as in regular neural network training. The gradient calculation with respect to $I$ can be derived using the chain rule of derivative. Specifically, the gradient calculation of $o_{batch}$ with respect to $Y_i$ can be derived as

$$\frac{\partial o_{batch}}{Y_i} = \frac{1}{b} \frac{\partial o_i}{Y_i} \ \text{ for } i = 1, \cdots, b. \tag{9}$$

Subsequently, the gradient calculation of $o_{batch}$ with respect to $I$ over a batch of training samples can be derived as

$$\frac{\partial o_{batch}}{\partial I} = \frac{1}{b} \sum_{j=1}^{b} \frac{\partial o_j}{\partial Y_j} \frac{\partial Y_j}{\partial I}$$

$$= [\frac{1}{b} \sum_{j=1}^{b} \frac{\partial o_j}{\partial Y_j[0]} \frac{\partial f(X_j, I[0])}{\partial I[0]}, \cdots, \frac{1}{b} \sum_{j=1}^{b} \frac{\partial o_j}{\partial Y_j[m-1]} \frac{\partial f(X_j, I[m-1])}{\partial I[m-1]}]. \tag{10}$$

Leveraging the gradient calculations in (9) and (10), the subproblem in (6) can be therefore solved by using gradient-descent-based algorithms. The outputs from the initial-training step include the optimized selection set $I_R$ and the corresponding reconstruction network parameters $\Theta_R$. As the entries of $I_R$ are continuous over interval $(-\infty, \infty)$, one needs further convert $I_R$ to an integer selection set $I_{opt}$ as shown in (7). To compensate for the accuracy loss due to rounding, the reconstruction network shall be further fine tuned in the fine-tuning step while keeping $I_{opt}$ fixed as shown in (7).

The entire algorithm of co-training a information decoding network $N$ and a selective sensing operator $S$ is summarized in **Algorithm** 1 in the appendix.

### 3.4 Image Selective Sensing and Reconstruction

In the rest of the paper, we take image as the sensing modality and reconstruction as the information decoding task to demonstrate the first proof-of-concept of selective sensing. The prior work discussed in subsection 2.3 shows that neural network models can be trained to directly approximate the inverse mapping of compressive sensing to perform the reconstruction. Therefore, we hypothesize that there exists a direct mapping from the selective sensed (nonuniformly subsampled) domain to the original image domain, and such a mapping can be well approximated by a neural network co-trained with the selective sensing operator. Furthermore, we hypothesize that the existing image compressive sensing reconstruction networks can be also used for image selective sensing reconstruction.

Based on our hypotheses, we use the loss function $l(\hat{X}, X)$ as the objective function in (2), where $l$ is a function that measures the distance between $\hat{X}$ and $X$, *e.g.* a mean-square-error function, and $\hat{X}$ is the output ($Z$ in (8)) of the information decoding network $N$. As such, $N$ is trained to directly reconstruct the original image from the selective sensing measurement as

$$ X \xrightarrow[\text{Sensing}]{S'} Y \xrightarrow[\text{Reconstruction}]{N} \hat{X}. \tag{11} $$

We refer to the image selective sensing and reconstruction frameworks composed in such way as SS+Net.

To evaluate the performance of SS+Net, we compare it against compressive sensing and uniform subsampling counterparts referred to as CS+Net and US+Net, respectively. CS+Net and US+Net use same reconstruction networks but replace the selective sensing operator in SS+Net with a Gaussian random sensing matrix[2] and a uniform subsampling operator[3], respectively. Additionally, we set all the hyper-parameters in SS+Net, CS+Net, and US+Net to be the same during the training for a fair comparison. The purpose of using CS+Net and US+Net as the reference methods is to reveal the true impact of selective sensing on compressive information acquisition in comparison to the compressive sensing and uniform subsampling counterparts.

## 4 Experiments

### 4.1 Experiment Setup

We conduct experiments on two datasets with two different reconstruction networks at the measurement/subsampling rates ranging from 0.03125 to 0.25 (corresponding to the dimensionality reduction ratios of 32-4x). The first dataset is CIFAR10 (Krizhevsky et al., 2009). The second dataset is composed in the same way as illustrated in Xu et al. (2018), which has 228,688 training samples and 867 testing samples. The testing samples are the non-overlapped image patches from Set5 (Bevilacqua et al., 2012) and Set14 (Zeyde et al., 2010). All the samples are of size 64x64 with three color channels(RGB). The sensing is performed channel-wise in all the experiments, i.e., for each framework of SS-Net, CS-Net and US-Net, each sample is first reshaped to three 4096-dimensional vectors corresponding to the three color channels. Subsequently, three sensing operators corresponding to the three color channels are used to sense the three vectors of each sample, respectively. The sensed measurements from the three color channels are grouped together and then fed into the reconstruction network. Prior to the training, 5% of the training samples are randomly selected and separated out as the validation set.

The two reconstruction networks we experimented with are DCNet and ReconNet (Kulkarni et al., 2016). DCNet has the same network structure as the generator network of DCGAN (Radford et al., 2015). We made some minor but necessary modifications to the structure of DCNet and ReconNet in order to perform image reconstruction and speed up the training. The modification details are summarized in subsection A.2 in the Appendix.

---

[2]The majority of existing data-driven image compressive sensing reconstruction methods (Kulkarni et al., 2016; Bora et al., 2017; Metzler et al., 2017; Xu et al., 2018; Zhang & Ghanem, 2018; Yao et al., 2019; Van Veen et al., 2018) use a fixed Gaussian random sensing matrix to perform the sensing.

[3]Assume the sampling rate is $\frac{m}{n}$, an input vector of length $n$ is equally divided into $m$ segments and the central entry of each segment is sampled as a measurement

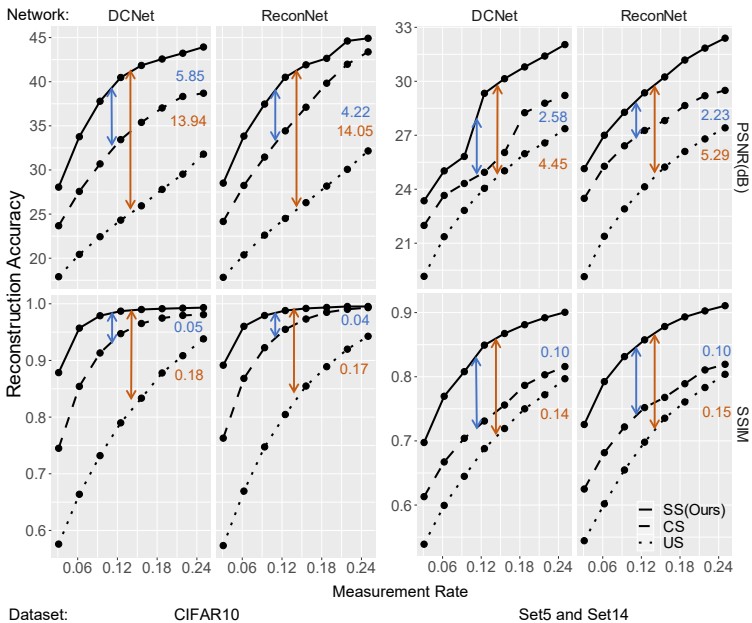

Figure 1: Comparison of information acquisition performance among selective sensing (SS), compressive sensing (CS), and uniform subsampling (US) measured in PSNR and SSIM. The average PSNR and SSIM improvements of SS over CS and US across all eight measurement rates are annotated on the figure.

In the beginning of the training process of SS+Net, the selection index $I$ is randomly initialized with real values from the uniform distribution $U(0, n)$. We co-train the selective sensing operator and the reconstruction network for 300 iterations, of which the first 150 iterations are used for the initial-training step and the rest are used for the fine-tuning step. We use two different optimizers to optimize different components of SS+Net: An Adam optimizer with a learning rate of 0.001 is used to optimize reconstruction networks and a SGD optimizer with a learning rate of 100000 is used to optimize sensing operators. Using a high learning rate for training the selective sensing operators is because the gradient values with respect to the selection index $I$ (calculated with (4)) turns out to be orders of magnitude smaller (because adjacent pixels in natural images mostly have very close pixel values) than the the rest of the gradient values and the learning rate of 100000 performs well in the experiments. For the training of the CS+Net and US+Net counterparts, except that there is no optimizer for sensing operators, all the other experiment setups remain the same with SS+Net.

## 4.2 EXPERIMENT RESULTS

The reconstruction accuracy is measured as average reconstruction PSNR and SSIM. The experiment results of PSNR and SSIM are plotted in Figure 1. As shown in Figure 1, selective sensing achieves up to 44.92dB/0.9952 reconstruction PSNR/SSIM at the measurement rate of 0.25 (dimensionality reduction ratio of 4). Even at the low measurement rate of 0.03125 (dimensionality reduction ratio of 32), selective sensing still achieves at least 23.35dB/0.6975 reconstruction PSNR/SSIM. The experiment results validate our hypothesis that the direct mapping from the selective sensed domain to the original image domain can be well approximated by existing reconstruction neural networks co-trained with the selective sensing operator, and data-driven nonuniform subsampling can well preserve signal information under the presence of the co-trained information decoding neural network. Furthermore, the experiment results show that selective sensing consistently outperforms compressive sensing and uniform subsampling, especially at higher dimensionality reduction ratios. The average PSNR/SSIM improvement of selective sensing over compressive sensing and uniform subsampling across all the experiments is 3.73dB/0.07 and 9.43dB/0.16, respectively. As the only difference between SS+Net, CS+Net, and US+Net is the sensing operator used, the experiment results imply that selective sensing better preserves signal information than compressive sensing and

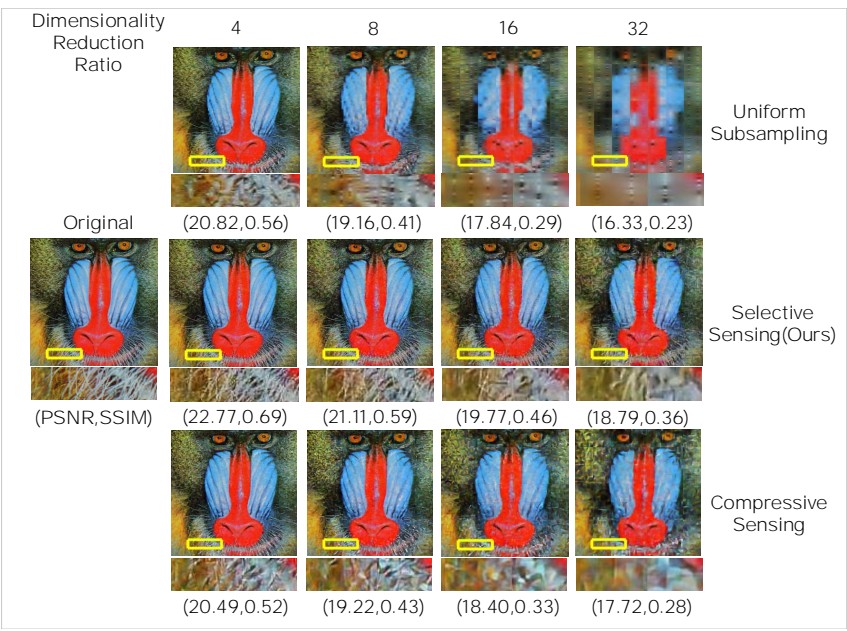

Figure 2: Visual reconstruction quality comparison among selective sensing, compressive sensing and uniform subsampling. The reconstruction network used is ReconNet, and the example image is from the Set14. Across all the dimensionality reduction ratios, selective sensing produces a sharper image with finer details presented, *e.g.* the beard and the skin textures.

uniform subsampling as a result of the co-optimization of the sensing and reconstruction stages. The detailed numerical PSNR and SSIM results are presented in Table 1, Table 2, Table 3 and Table 4 in the Appendix.

Compared with the images in CIFAR10, the images in Set5 and Set14 have more details, which makes the reconstruction inherently more difficult. We take one image from Set14 as an example to illustrate the visual reconstruction quality comparison. As shown in Figure 2, selective sensing reconstructs the image with finer and sharper details, such as the beard and the textures of the skin, than compressive sensing and uniform subsampling across all dimensionality reduction ratios. More visual reconstruction quality comparison showing the same results are illustrated in Figure 3, Figure 4 and Figure 5 in the Appendix. These visual reconstruction quality comparisons show strong evidence that selective sensing better preserves signal information than compressive sensing and uniform subsampling.

### 4.3 COMPARISON AGAINST RANDOM INDICES

To understand the effectiveness of the trained selection indices in selective sensing over randomly generated selection indices, we design a trivial random subsampling approach called RS+Net and compared it with SS+Net. The experimental setup for RS+Net is consistent with US+Net except that the selection indices of RS+Net are integers randomly generated from the uniform distribution $U(0, n)$. The experiments are performed on CIFAR10 at the dimensionality reduction ratios of 32 and 8 (measurement rates of 0.03125 and 0.125) with ReconNet. At each measurement rate, we randomly generate eight different sets of selection indices, based on which we train eight reconstruction networks from scratch, respectively, for RS+Net.

We observe that the final reconstruction accuracy of RS+Net on average is significantly lower than SS+Net counterparts. The average PSNR of RS+Net is 22.78 dB and 31.02 dB at measurement rates of 0.03125 and 0.125, which is 5.73 dB and 9.49 dB lower than the SS+Net counterparts, respectively. In addition, we observe a large variance in the final reconstruction accuracy of RS+Net across the eight sets of selection indices. Specifically, at the measurement rate of 0.03125 and 0.125, the gap between the maximum and the minimum PSNRs is 0.59 dB and 0.54dB, and the standard deviation of PSNRs is 0.19 dB and 0.20 dB, respectively. The detailed results are summarized in Ta-

ble 5 in the Appendix. Furthermore, to illustrate the superiority of the co-optimized selection indices in selective sensing over random selection indices, we also train two reconstruction networks from scratch at the measurement rates of 0.03125 and 0.0125 with the corresponding pre-trained selection indices from SS+Net remaining fixed in the training process, respectively. The final reconstruction accuracy is 28.49 dB and 40.47 dB at the measurement of 0.03125 and 0.0125, which is 5.71 dB and 9.45 dB higher than the original RS+Net counterparts using random selection indices, respectively.

These results indicate that random selection indices are insufficient for retaining information of interest with respect to the subsequent reconstruction network. Differently, learned indices from co-training a selective sensing operator with a subsequent reconstruction network can significantly improve the selective sensing performance in terms of retaining the information of interest for high-accuracy reconstruction. The visualization and comparison of learned (co-trained) and random selection masks are shown in Figure 6, 7, and 8 in subsection A.3 of the Appendix.

### 4.4 DISCUSSION ON NONCONVEXITY AND INCOHERENCE

From the optimization perspective, both the original problem 2 and the relaxed problem 6 are non-convex problems. Thus, it is not guaranteed that the gradient-descent-based algorithm 1 can find the globally optimal solution of the problem 2. Additionally, it is known that deep neural network models are typically non-convex with a great number of local minima (Goodfellow et al., 2016). The gap between local and global minima remains an open topic of research. Similarly, our approach leverages the gradient-descent algorithm and relaxed problem 6 to find the local minimum of problem 2 that are empirically proven to have outstanding performance. Specifically, the experiment results show that the proposed algorithm 1 can consistently find a local optimum of the indices $I$ that has significantly better sensing performance than random indices and random Gaussian sensing matrices in terms of the reconstruction accuracy across different datasets and compression ratios.

It should be also noted that selective sensing, similar to many data-driven compressive sensing methods that co-train a sensing matrix with a reconstruction network(Mousavi et al., 2017; 2018; Lohit et al., 2018), does not explicitly require the incoherence of sensing matrices nor sparsity basis. Previous studies have empirically shown that these data-driven compressive sensing methods can achieve significantly higher reconstruction performance compared to conventional model-based methods(Li et al., 2013; Dong et al., 2014) that have the incoherence and sparsity requirements. Similarly, our experiment results on various datasets, reconstruction networks, and measurement rates also show that selective sensing consistently outperforms compressive sensing and uniform subsampling counterparts. Furthermore, compared to the data-driven compressive sensing method in Lohit et al. (2018), selective sensing can achieve 6.02 dB/0.031 and 0.72 dB/0.007 higher reconstruction PSNR/SSIM on the CIFAR10 database with the same reconstruction network Kulkarni et al. (2016) at the measurement rate of 0.125 and 0.25, respectively. These comparison results are the strong empirical evidence that selective sensing can well preserve signal information without the incoherence nor sparsity requirements.

## 5 CONCLUSION

In this paper, we propose a selective sensing framework that adopts the novel concept of data-driven nonuniform subsampling to perform on-sensor data dimensionality reduction in a computation-free fashion. Selective sensing adopts a co-optimization methodology to co-train a selective sensing operator with a subsequent information decoding neural network. The co-training of selective sensing is first formulated as a mixed-discrete-continuous optimization problem. By applying continuous interpolation and domain extension to the sensing index domain with quantization and fine-tuning techniques, we reformulate the problem into two continuous optimization subproblems that can be solved by gradient-descent-based algorithms. This is the key to enabling the co-training of the selective sensing operator with the subsequent information decoding neural network. The experiments on CIFAR10, Set5, and Set14 datasets show that the proposed selective sensing framework can achieve an average reconstruction accuracy improvement in terms of PSNR/SSIM of 3.73dB/0.07 and 9.43dB/0.16 over compressive sensing and uniform subsampling counterparts across the dimensionality reduction ratios of 4-32x, respectively. The computation-free nature of selective sensing makes it a highly suitable solution for performing compressive information acquisition on resource-constrained sensor devices or high-data-rate sensor devices dealing with high-dimensional signals.

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

# A APPENDIX

## A.1 SUMMARIZED ALGORITHM

---

**Algorithm 1** Main algorithm

---

**Input:** training samples $X_1, \cdots, X_N$, number of iterations $maxiters$ for initial-training step, batch size $b$

Initialize $I, \Theta$

**Initial-training**

**for** $iter = 1$ **to** $maxiters$ **do**

    **for** $batch = 1$ **to** $\frac{N}{b}$ **do**

        **Forward pass**

        Execute (8)

        **Backward pass**

        Using the BP algorithm to calculate the gradient with respect to $\Theta$

        Execute (9)

        Execute (10) to calculate the gradient with respect to $I$

        Optimize $I$ and $\Theta$ using the calculated gradients

    **end for**

**end for**

$I_R, \Theta_R = I, \Theta$.

Execute the rounding and modulo operations over entries of $I_R$ as in (7) to get $I_{opt}$.

**Fine-tuning**

Initialize $\Theta = \Theta_R$, further optimize $\Theta$ with the gradients calculated by the BP algorithm while keeping $I = I_{opt}$

**End**

$I_{opt}, \Theta_{opt} = I, \Theta$

---

## A.2 MORE EXPERIMENTS DETAILS

The code, datasets and pretrained models can be downloaded from:

`https://figshare.com/s/519a923fae8f386d7f5b.`

The experiments are conducted in parallel on four RTX 2080 Ti GPU cards. One training process(300 iterations) runs on one GPU card takes around 80 minutes.

We made following necessary modifications to the structure of DCNet and ReconNet in order to perform image reconstruction and speed up the training process. We modify the first layer of DCNet by increasing the number of input channels to be equal to the number of measurements in order to feed measurements into DCNet. As ReconNet is designed to reconstruct grayscale images, we made some modifications to the network structure of ReconNet in order to reconstruct color images. Specifically, we replace the first layer of ReconNet, which is a fully-connected layer, with three fully-connected layers in parallel to generate three feature map corresponding to three color channels. In addition, we modify the third and the sixth convolutional layers to have three output channels in order to maintain the dimensionality of the original image. The batch normalization layers are also added right behind each convolutional layer (except for the last convolutional layer which is the output layer) to accelerate the training.

## A.3   MORE EXPERIMENT RESULTS

Figure 6,Figure 7 and Figure 8 shows that sampling points in learned masks are more evenly distributed on the whole image and less clustered at specific locations across all measurement rates than random masks. Consequently, the measurements from selective sensing are less noisy but more informative than random sensing. For instance, the outlines of objects in the measurements sensed by selective sensing are more visually comprehensible than random sensing, which provides a possible visual explanation on the effectiveness of selective sensing over random subsampling.

Table 1: Reconstruction performance comparison on CIFAR10 using DCNet as the reconstruction network.

| Metrics | | PSNR (dB) | | | | |
|---|---|---|---|---|---|---|
| Sensing operator | | SS | CS | US | SS over CS | SS over US |
| Measurement rate | 0.03125 | 28.055576 | 23.678496 | 17.903165 | 4.3770796 | 10.152411 |
| | 0.0625 | 33.76353 | 27.567673 | 20.432642 | 6.1958566 | 13.330888 |
| | 0.09375 | 37.788893 | 30.687367 | 22.436043 | 7.1015266 | 15.352851 |
| | 0.125 | 40.4781 | 33.434791 | 24.314229 | 7.043309 | 16.163872 |
| | 0.15625 | 41.843324 | 35.391706 | 25.934857 | 6.4516183 | 15.908467 |
| | 0.1875 | 42.571765 | 37.023333 | 27.796602 | 5.5484323 | 14.775163 |
| | 0.21875 | 43.227788 | 38.326256 | 29.525659 | 4.9015314 | 13.702128 |
| | 0.25 | 43.928165 | 38.693752 | 31.793804 | 5.2344128 | 12.134361 |
| Metrics | | SSIM | | | | |
| Sensing operator | | SS | CS | US | SS over CS | SS over US |
| Measurement rate | 0.03125 | 0.8785 | 0.745 | 0.5759 | 0.1335 | 0.3026 |
| | 0.0625 | 0.957 | 0.8542 | 0.6638 | 0.1028 | 0.2932 |
| | 0.09375 | 0.9789 | 0.9132 | 0.732 | 0.0657 | 0.2469 |
| | 0.125 | 0.987 | 0.9473 | 0.7899 | 0.0397 | 0.1971 |
| | 0.15625 | 0.9899 | 0.9652 | 0.8334 | 0.0247 | 0.1565 |
| | 0.1875 | 0.9913 | 0.9747 | 0.8779 | 0.0166 | 0.1134 |
| | 0.21875 | 0.9923 | 0.9795 | 0.9084 | 0.0128 | 0.0839 |
| | 0.25 | 0.9932 | 0.9809 | 0.9381 | 0.0123 | 0.0551 |

Table 2: Reconstruction performance comparison on CIFAR10 using ReconNet as the reconstruction network.

| Metrics | | PSNR (dB) | | | | |
|---|---|---|---|---|---|---|
| Sensing operator | | SS | CS | US | SS over CS | SS over US |
| | 0.03125 | 28.510575 | 24.162001 | 17.833156 | 4.3485742 | 10.677419 |
| | 0.0625 | 33.835395 | 28.244744 | 20.37912 | 5.5906511 | 13.456275 |
| | 0.09375 | 37.465897 | 31.460441 | 22.615043 | 6.005456 | 14.850854 |
| | 0.125 | 40.505386 | 34.428422 | 24.507658 | 6.0769649 | 15.997729 |
| | 0.15625 | 41.91067 | 37.112229 | 26.295812 | 4.7984413 | 15.614858 |
| Measurement rate | 0.1875 | 42.64523 | 39.830833 | 28.173974 | 2.8143966 | 14.471256 |
| | 0.21875 | 44.613196 | 41.977892 | 30.068614 | 2.6353032 | 14.544581 |
| | 0.25 | 44.922706 | 43.377697 | 32.157417 | 1.5450094 | 12.765289 |
| Metrics | | SSIM | | | | |
| Sensing operator | | SS | CS | US | SS over CS | SS over US |
| | 0.03125 | 0.8916 | 0.7629 | 0.5733 | 0.1287 | 0.3183 |
| | 0.0625 | 0.9602 | 0.8683 | 0.6694 | 0.0919 | 0.2908 |
| | 0.09375 | 0.9793 | 0.9225 | 0.7474 | 0.0568 | 0.2319 |
| | 0.125 | 0.988 | 0.9549 | 0.8046 | 0.0331 | 0.1834 |
| | 0.15625 | 0.9918 | 0.9731 | 0.8548 | 0.0187 | 0.137 |
| Measurement rate | 0.1875 | 0.9935 | 0.9849 | 0.889 | 0.0086 | 0.1045 |
| | 0.21875 | 0.9952 | 0.9903 | 0.9199 | 0.0049 | 0.0753 |
| | 0.25 | 0.9952 | 0.9932 | 0.9427 | 0.002 | 0.0525 |

Table 3: Reconstruction performance comparison on Set5 and Set14 using DCNet as the reconstruction network.

| Metrics | | PSNR (dB) | | | | |
|---|---|---|---|---|---|---|
| Sensing operator | | SS | CS | US | SS over CS | SS over US |
| Measurement rate | 0.03125 | 23.356654 | 21.988134 | 19.158458 | 1.3685191 | 4.1981953 |
| | 0.0625 | 25.020666 | 23.655472 | 21.360483 | 1.3651944 | 3.6601834 |
| | 0.09375 | 25.821571 | 24.320524 | 22.822223 | 1.5010468 | 2.9993483 |
| | 0.125 | 29.330996 | 24.941122 | 24.049497 | 4.3898736 | 5.2814991 |
| | 0.15625 | 30.142491 | 26.041837 | 25.023534 | 4.1006538 | 5.1189574 |
| | 0.1875 | 30.798615 | 28.254278 | 25.973201 | 2.5443367 | 4.8254134 |
| | 0.21875 | 31.402867 | 28.782112 | 26.572408 | 2.6207543 | 4.8304582 |
| | 0.25 | 32.040132 | 29.214131 | 27.364247 | 2.8260015 | 4.6758851 |
| Metrics | | SSIM | | | | |
| Sensing operator | | SS | CS | US | SS over CS | SS over US |
| Measurement rate | 0.03125 | 0.6975 | 0.6131 | 0.539 | 0.0844 | 0.1585 |
| | 0.0625 | 0.7696 | 0.6672 | 0.5995 | 0.1024 | 0.1701 |
| | 0.09375 | 0.8079 | 0.7041 | 0.6448 | 0.1038 | 0.1631 |
| | 0.125 | 0.8493 | 0.731 | 0.688 | 0.1183 | 0.1613 |
| | 0.15625 | 0.8674 | 0.7559 | 0.7193 | 0.1115 | 0.1481 |
| | 0.1875 | 0.8811 | 0.7866 | 0.75 | 0.0945 | 0.1311 |
| | 0.21875 | 0.8919 | 0.803 | 0.7719 | 0.0889 | 0.12 |
| | 0.25 | 0.9005 | 0.8159 | 0.7969 | 0.0846 | 0.1036 |

Table 4: Reconstruction performance comparison on Set5 and Set14 using ReconNet as the reconstruction network.

| Metrics | | PSNR (dB) | | | | |
|---------|---|-----|-----|-----|-----------|-----------|
| Sensing operator | | SS | CS | US | SS over CS | SS over US |
| | 0.03125 | 25.14523 | 23.492886 | 19.141701 | 1.652344 | 6.0035286 |
| | 0.0625 | 27.002097 | 25.27621 | 21.385219 | 1.725887 | 5.6168772 |
| | 0.09375 | 28.276923 | 26.41306 | 22.907161 | 1.8638628 | 5.3697623 |
| | 0.125 | 29.357467 | 27.2674 | 24.135962 | 2.0900672 | 5.2215052 |
| | 0.15625 | 30.240035 | 27.814269 | 25.232525 | 2.4257659 | 5.0075104 |
| Measurement rate | 0.1875 | 31.18125 | 28.651692 | 26.099135 | 2.5295576 | 5.0821148 |
| | 0.21875 | 31.844453 | 29.195588 | 26.796003 | 2.6488653 | 5.0484498 |
| | 0.25 | 32.397313 | 29.493561 | 27.412122 | 2.9037527 | 4.9851912 |
| Metrics | | SSIM | | | | |
| Sensing operator | | SS | CS | US | SS over CS | SS over US |
| | 0.03125 | 0.7256 | 0.625 | 0.5444 | 0.1006 | 0.1812 |
| | 0.0625 | 0.7924 | 0.6816 | 0.6019 | 0.1108 | 0.1905 |
| | 0.09375 | 0.8314 | 0.7218 | 0.6547 | 0.1096 | 0.1767 |
| | 0.125 | 0.8575 | 0.7519 | 0.698 | 0.1056 | 0.1595 |
| | 0.15625 | 0.8783 | 0.7678 | 0.7354 | 0.1105 | 0.1429 |
| Measurement rate | 0.1875 | 0.8932 | 0.7891 | 0.7607 | 0.1041 | 0.1325 |
| | 0.21875 | 0.9026 | 0.8106 | 0.7832 | 0.092 | 0.1194 |
| | 0.25 | 0.9107 | 0.8194 | 0.8038 | 0.0913 | 0.1069 |

Table 5: Summary of reconstruction accuracy results of RS+Net on CIFAR10 dataset. Metrics: PSNR(dB)/SSIM

| Measurement rate | Mean | Min | Max | Standard deviation |
|------------------|------|-----|-----|--------------------|
| 0.03125 | 22.78/0.74 | 22.57/0.73 | 23.16/0.75 | 0.19/0.006 |
| 0.125 | 31.02/0.93 | 30.71/0.93 | 31.25/0.94 | 0.20/0.0018 |

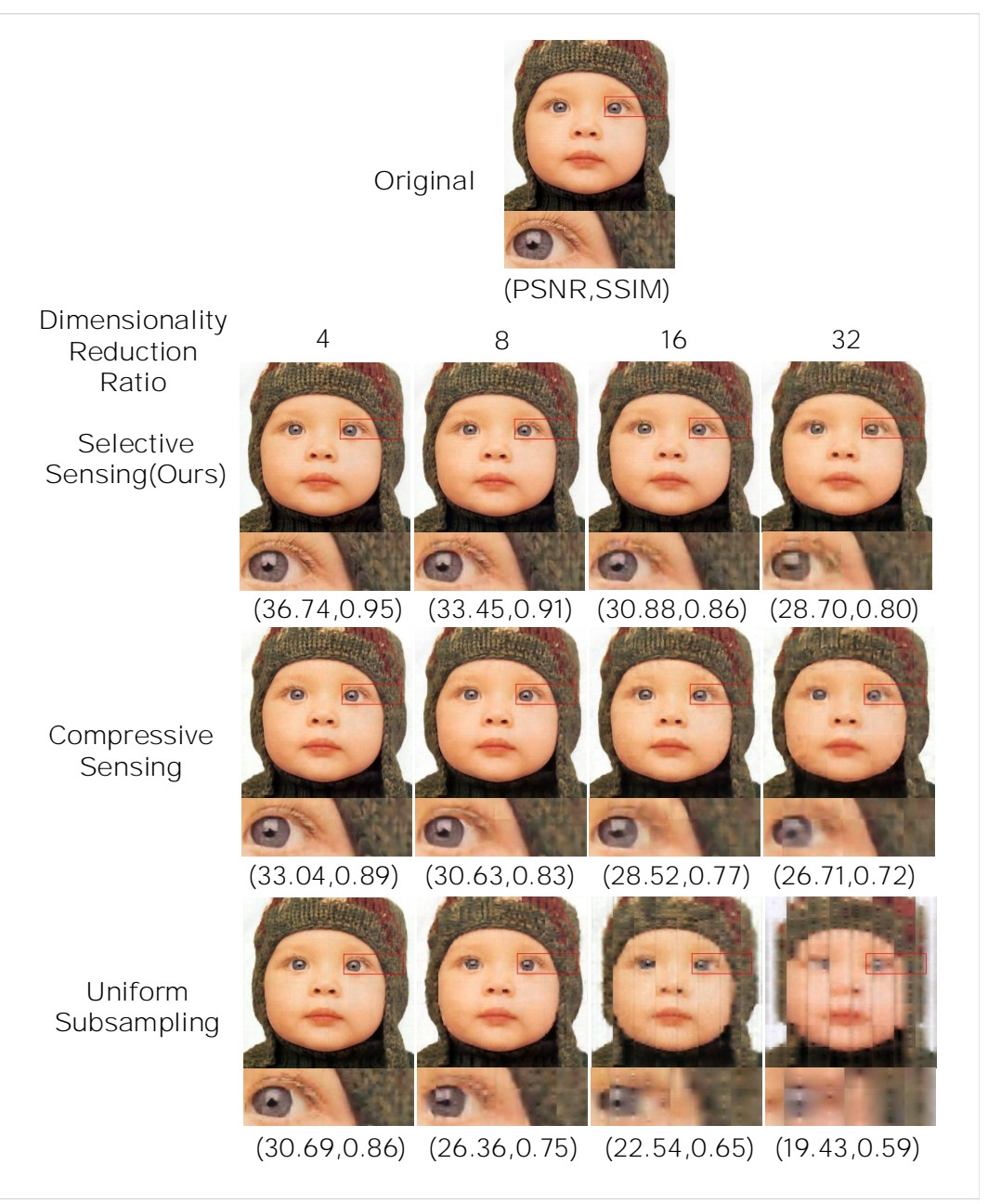

Figure 3: Visual reconstruction quality comparison among selective sensing, compressive sensing and uniform subsampling. The reconstruction network used is ReconNet, and the example image is from the Set5 dataset. Across all the compression ratios, selective sensing produces a sharper image with finer details presented, *e.g.* the eyelashes and the textures of the hat.

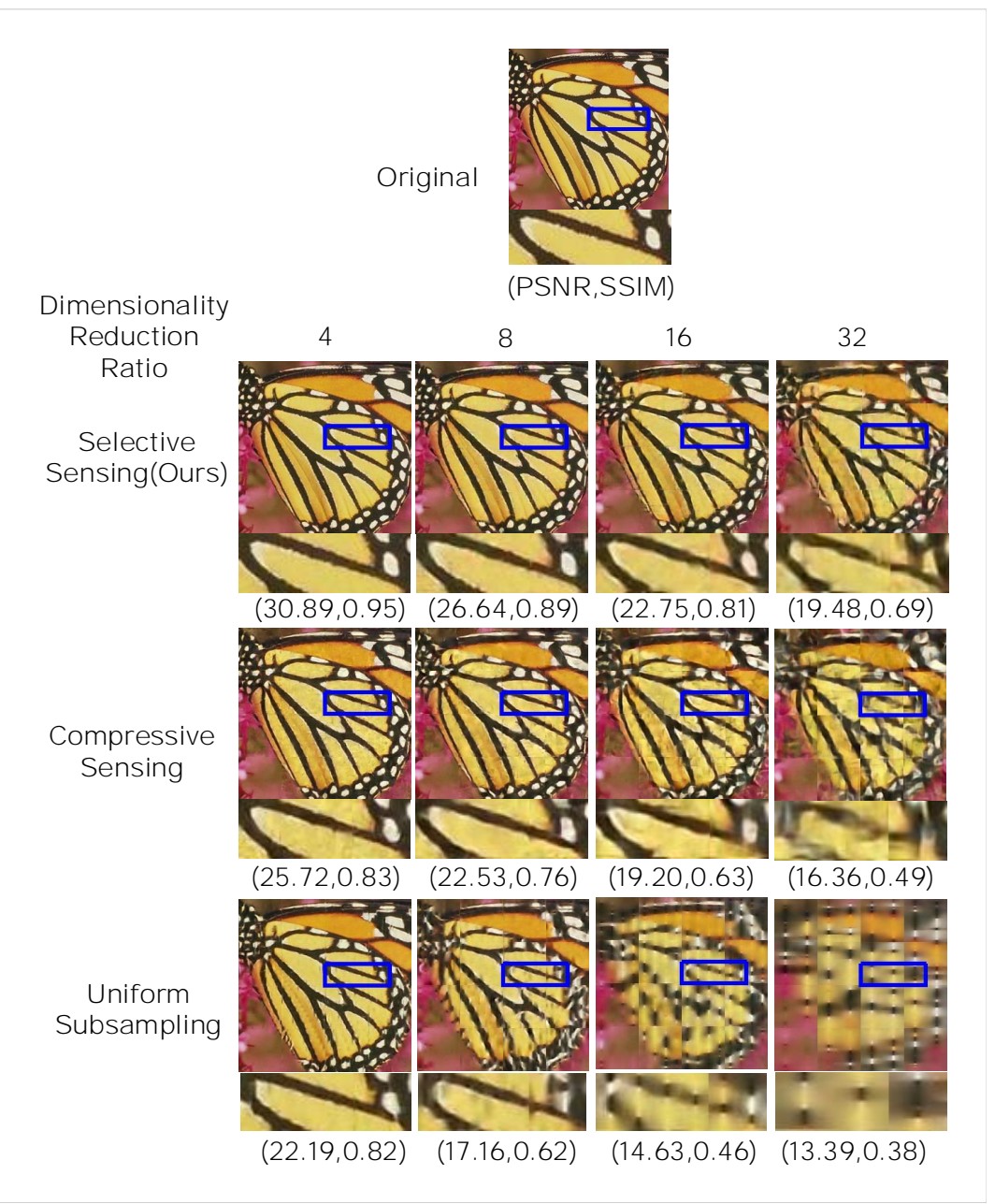

Figure 4: Visual reconstruction quality comparison among selective sensing, compressive sensing and uniform subsampling. The reconstruction network used is ReconNet, and the example image is from the Set5 dataset. Across all the compression ratios, selective sensing produces a sharper image with finer details presented, *e.g.* the textures on the wings.

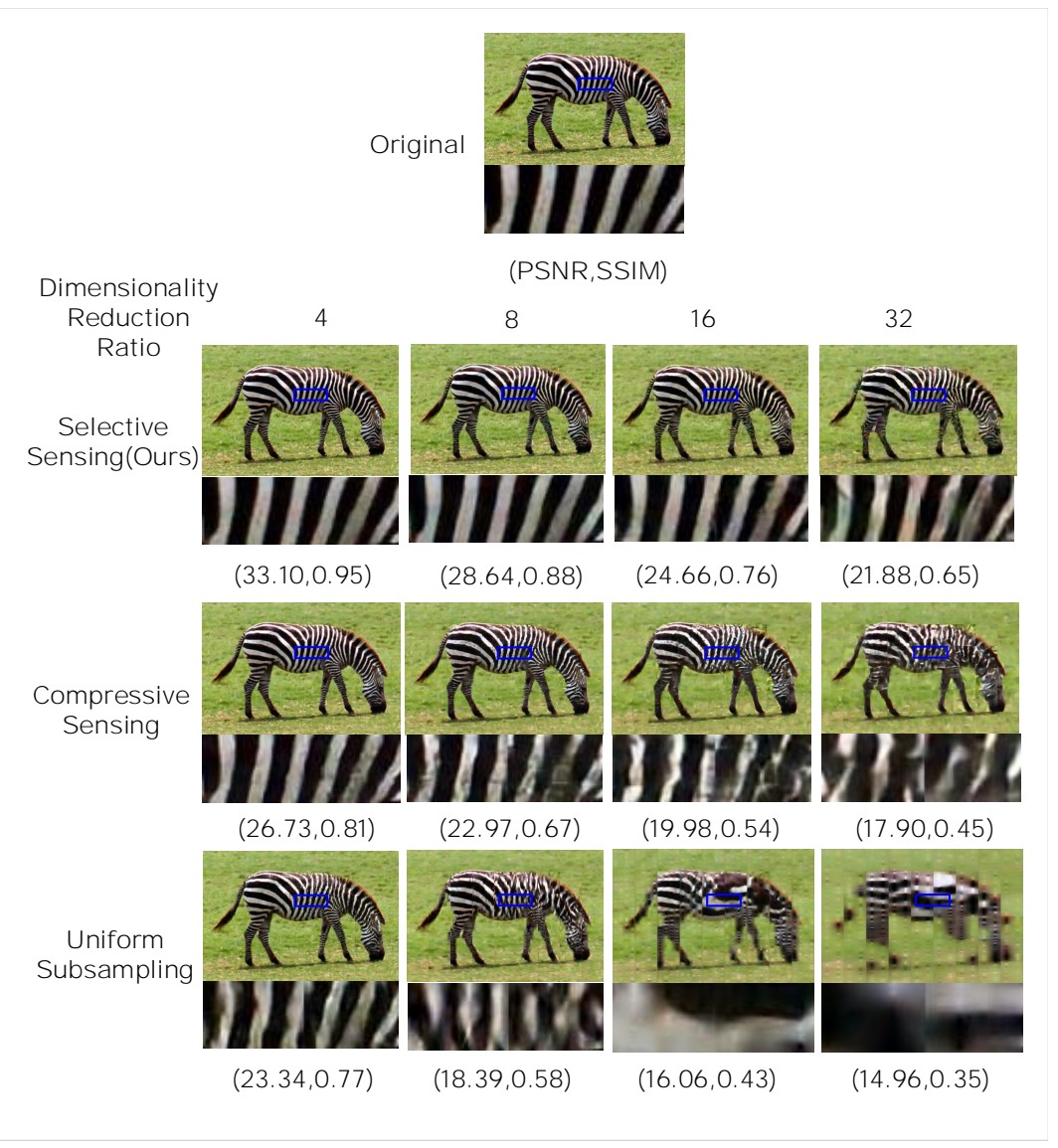

Figure 5: Visual reconstruction quality comparison among selective sensing, compressive sensing and uniform subsampling. The reconstruction network used is ReconNet, and the example image is from the Set14 dataset. Across all the compression ratios, selective sensing produces a sharper image with finer details presented, *e.g.* the edges of the stripes.

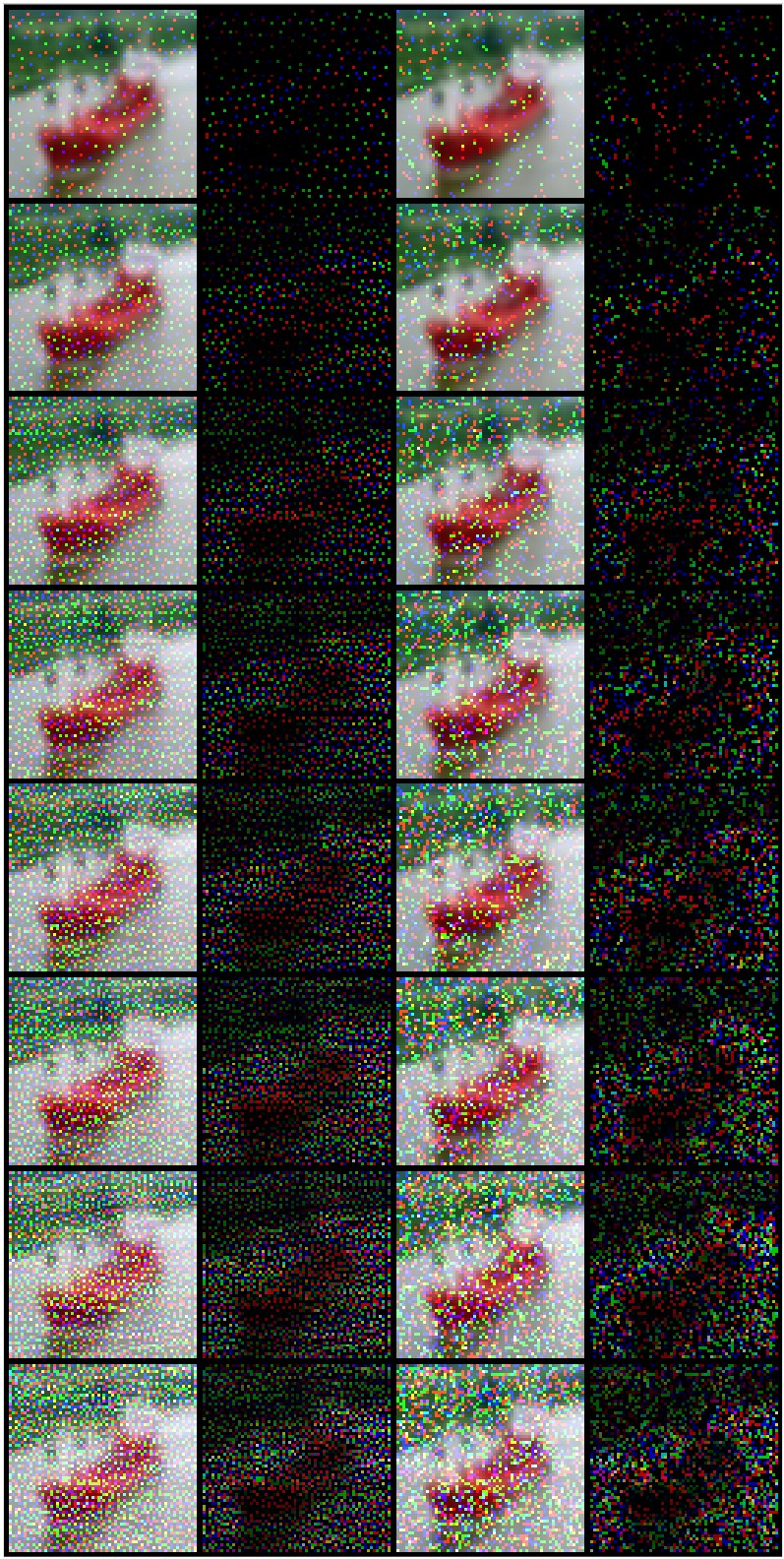

Figure 6: Visual comparison between learned and random selection masks. Top to bottom: masks at the measurement rates from 0.03125 to 0.25. Left to right: learned selection masks of selective sensing, selective sensing measurements, random selection masks, and randomly selected measurements. The masks and measurements are plotted based on color channels.

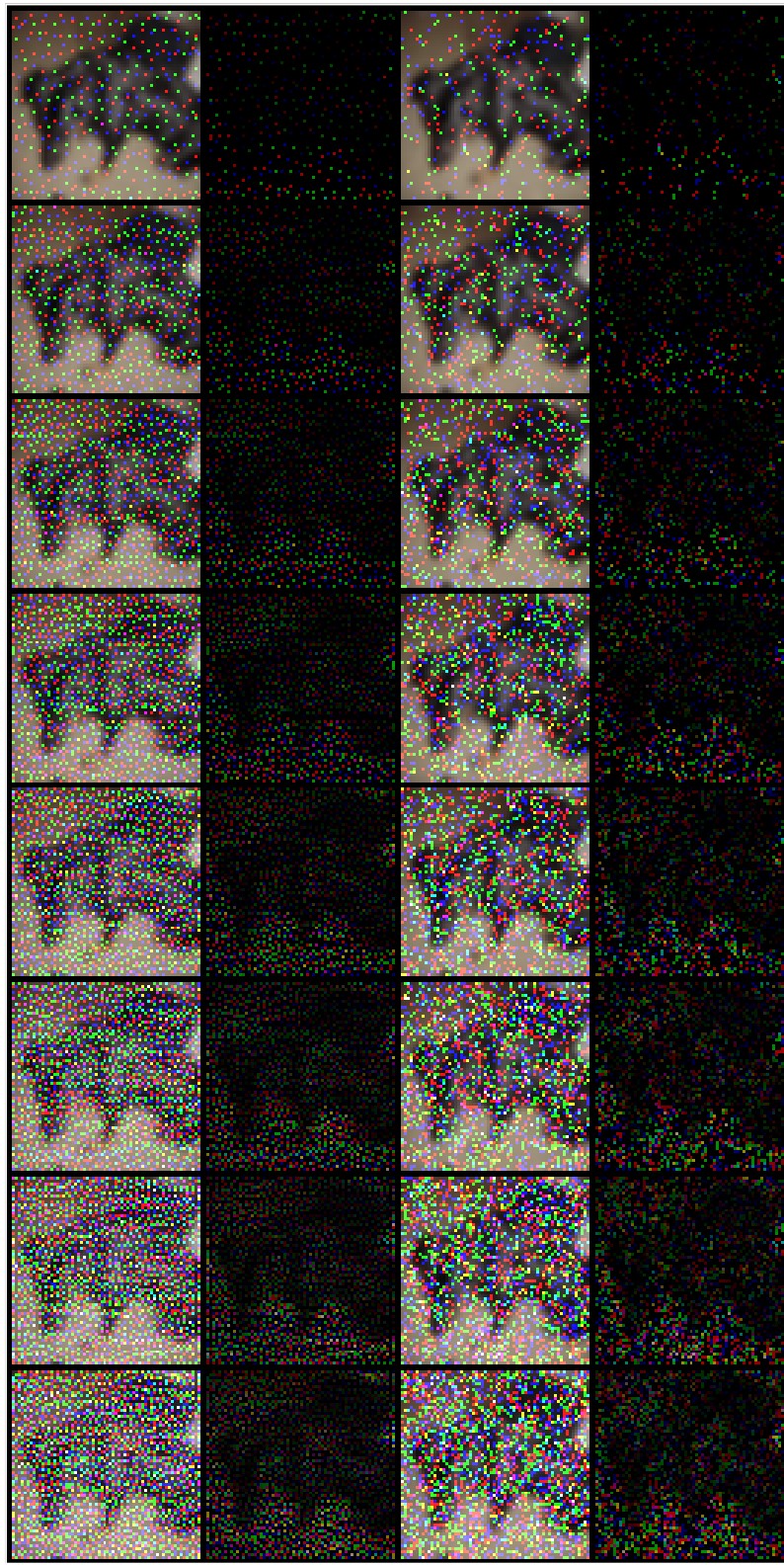

Figure 7: Visual comparison between learned and random selection masks. Top to bottom: masks at the measurement rates from 0.03125 to 0.25. Left to right: learned selection masks of selective sensing, selective sensing measurements, random selection masks, and randomly selected measurements. The masks and measurements are plotted based on color channels.

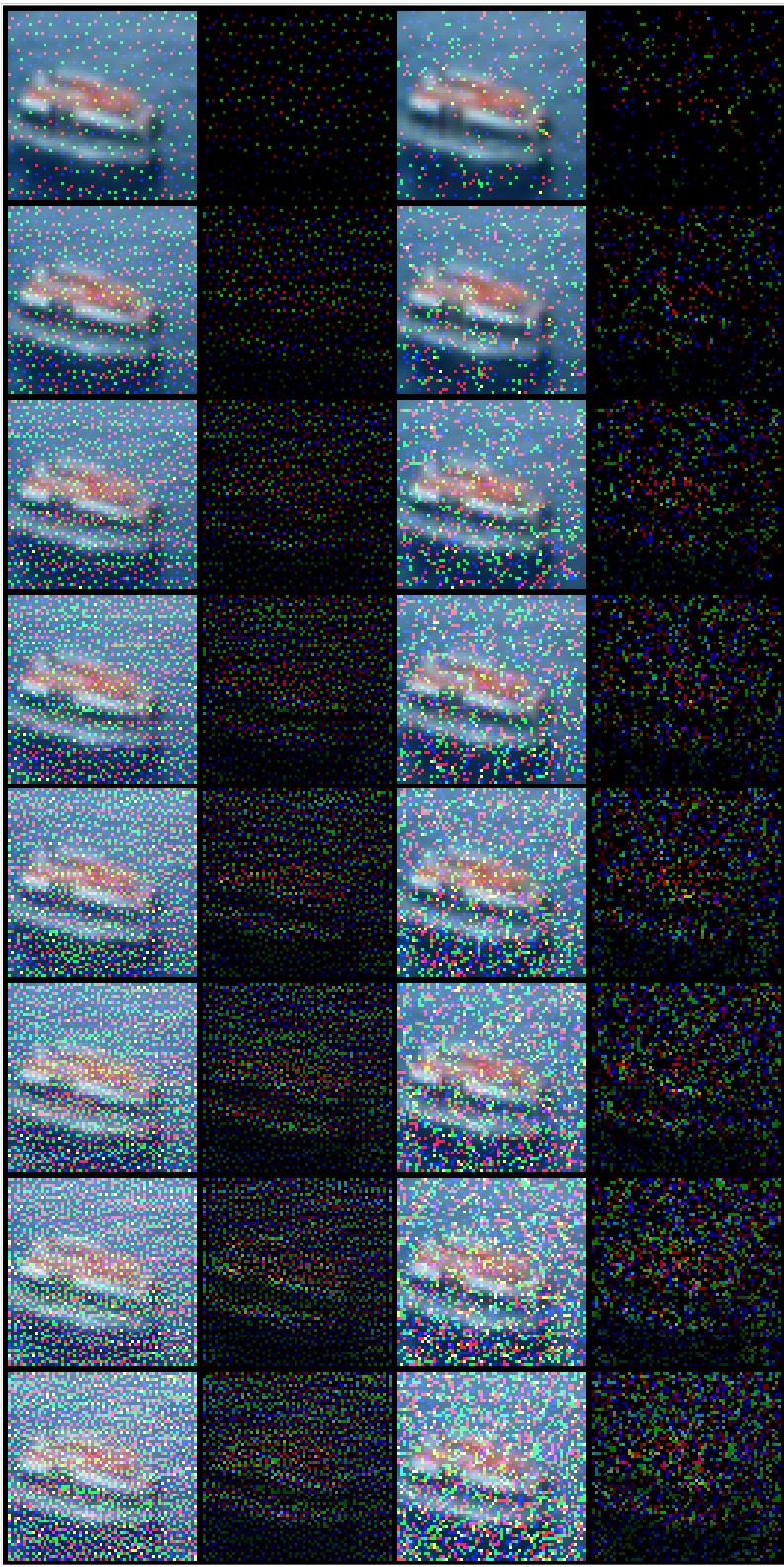

Figure 8: Visual comparison between learned and random selection masks. Top to bottom: masks at the measurement rates from 0.03125 to 0.25. Left to right: learned selection masks of selective sensing, selective sensing measurements, random selection masks, and randomly selected measurements. The masks and measurements are plotted based on color channels.

