# OpenReview forum: "Selective Sensing: A Data-driven Nonuniform Subsampling Approach for Computation-free On-Sensor Data Dimensionality Reduction"
_ICLR.cc/2021/Conference — Reject_

### Official Review · AnonReviewer2 · 2020-10-28
**Anonymous review of selective sensing paper**

**Rating:** 4
**Confidence:** 5

**Review:**

Summary: The paper proposes a framework for jointly optimizing a selective sensing operator and a neural network for reconstruction. The motivation is to alleviate the quadratic (in dimensions) complexity associated with standard compressive sensing by using a dimension-free selective sensing approach, which can be jointly optimized with the decoder while guaranteeing that the relevant information is preserved. As this formulation yields a mixed discrete-continuous optimization, the authors propose a standard relaxation of the discrete constraints using interpolation.

Strengths: The paper is generally well-written and organized and easy to follow. The idea of jointly optimizing the sensing and the decoder is interesting, although not really that novel.

Weaknesses: While the work has some merit, the development of ideas is lacking in many respects. Here are few major comments I have:
- The paper incorrectly overstressed the limitation of compressive sensing due to the computational complexity of applying a matrix, say from a Gaussian random ensemble, to the signal of interest. In many applications, applying such matrices follows from the structure of the sensing device. For example, applying a Fourier matrix, which come with provable guarantees, can be done using a lens! In other words, physics does it for you for free. In other cases, a mask amounts to applying a Hadamard matrix to your image.
- At a fundamental level, I am questioning the generality of the proposed approach which uses selective sampling. More specifically, it is well recognized that compressive reconstruction from a small number of measurements requires the incoherence of the sensing matrix and the sparsifying basis. This is known not to hold with the sparse matrices associated with selective sampling. The paper does not even explain any of these issues and limitations, and does not investigate it.
- The main motivation is to reduce the computational complexity of sensing. However, the paper does not provide any comparisons to the state-of-the-art concerning computational gains. While the joint optimization of the selector and the decoder is done during the training phase, in practice they will need to be re-trained for different datasets and scenarios, which is not necessarily going to save on computation.
- There is a large body of work on non-uniform compressive sampling that is not discussed in this work.
- Autoencoders can also be trained to obtain compressed representations that preserve information. This is a well-studied area that the paper fails to discuss.

Given the above, I am unable to recommend this paper for publication. For me, it is a clear reject.

---

> ### Author Response · Authors · 2020-11-24
> **Response to the comments(part 1)**
>
> We thank the reviewer for reviewing the paper and providing constructive feedback.
>
> Reviewer’s comment: The paper incorrectly overstressed the limitation of ... a mask amounts to applying a Hadamard matrix to your image.
>
> Response: We agree that custom hardware can be used to implement the compressive sensing process in the analog domain. We respectfully argue that such custom hardware implementation inevitably increases the cost of the sensor. In addition, such custom hardware implementation is often specific to the sensor design, thereby cannot be generally applied to other sensors or applications. Differently, the selective sensing method is developed as a much more general low-cost sensing method mostly for resource-constrained, low-cost sensors. In this paper, we use image selective sensing just to provide the first proof-of-concept. Due to the computation-free nature of selective sensing, our method not only can provide a significant saving on the computational cost for those applications that must implement compressive/selective sensing in the digital domain (such as IoT sensors for time series) but also can reduce the cost of the custom hardware implementation required for those applications that implement the sensing in the analog domain, (such as the single-pixel camera).  Because the selective sensing operator (nonuniform subsampling) can be simply implemented as a static mask in the analog domain. We revised the discussion on the necessity of selective sensing in section 1, paragraph 1 of our revised paper, to clarify this point to all the readers.
>
> Reviewer's comment: At a fundamental level, I am questioning the generality of ... and does not investigate it.
>
> Response: It should be also noted that selective sensing, similar to many data-driven compressive sensing methods that co-train a sensing matrix with a reconstruction network[5,6,7], does not explicitly require the incoherence of sensing matrices nor sparsity basis. Previous studies have empirically shown that these data-driven compressive sensing methods can achieve significantly higher reconstruction performance compared to conventional model-based methods[8,9] that have the incoherence and sparsity requirements. Similarly, our experiment results on various datasets, reconstruction networks, and measurement rates show that selective sensing consistently outperforms compressive sensing and uniform subsampling counterparts. Furthermore, compared to the data-driven compressive sensing method in [7], selective sensing can achieve 6.02 dB/0.031 and 0.72 dB/0.007 higher reconstruction PSNR/SSIM on the CIFAR10 database with the same reconstruction network[11] at the measurement rate of 0.125 and 0.25, respectively. These comparison results are the strong empirical evidence that selective sensing can well preserve signal information without the incoherence nor sparsity requirements.
>
> Reviewer's comment: The main motivation is to reduce the computational complexity of sensing. However, the paper does not provide any comparisons to the state-of-the-art concerning computational gains.
>
> Response: In Section 2.2 of the paper, we explain based on theoretical analysis that the computational complexity of selective sensing is $O(1)$, and we do compare the computational complexity of selective sensing with existing approaches[1,2]. The accurate evaluation of the computation gains  will require actual hardware implementation, which is beyond the scope of this paper that aims to provide the first proof-of-concept for selective sensing from the algorithm perspective. We will defer it to future work.
>
> Reviewer's comment: While the joint optimization of the selector and the decoder is done during the training phase, in practice they will need to be re-trained for different datasets and scenarios, which is not necessarily going to save on computation.
>
> Response: This is an unfortunate misunderstanding. We respectfully argue that selective sensing is a computation-free on-sensor data-dimensionality reduction technique. In other words, the computation-free property is claimed solely by the sensing part of selective sensing. The co-training of sensing operator(selector) and reconstruction network(decoder) of selective sensing is a one-time effort. Once the sensing operator is deployed to sensor devices, no further computation to update the sensing operator is needed. Based on the above illustration, we believe that the computation-free property of selective sensing is clear and justified.
>
> Reviewer's comment: There is a large body of work on non-uniform compressive sampling that is not discussed in this work.
>
> We add a new discussion on the differences between existing nonuniform subsampling approaches and selective sensing in section 2.1 of our revised paper to clarify this to all readers.

---

> ### Author Response · Authors · 2020-11-24
> **Response to the comments(part 2)**
>
> Reviewer's comment: Autoencoders can also be trained to obtain compressed representations that preserve information. This is a well-studied area that the paper fails to discuss.
>
> Response: Co-trained signal sensing and reconstruction frameworks can be viewed as a specific type of autoencoders[10]. The main difference between such frameworks and a general autoencoder model is that the sensing (encoder) part of such frameworks must be implemented on sensors for on-sensor data dimensionality reduction. Therefore, the computation complexity of the encoder has to be extremely low in order to be affordable for sensor devices. We add a new discussion on the differences between autoencoders and selective sensing in section 2.3 of our revised paper to clarify this to all readers.
>
> [1]Zhao, Wenfeng, et al. "On-chip neural data compression based on compressed sensing with sparse sensing matrices." IEEE transactions on biomedical circuits and systems 12.1 (2018): 242-254.
>
> [2]Hong, Tao, et al. "Optimized structured sparse sensing matrices for compressive sensing." Signal Processing 159 (2019): 119-129.
>
> [3]Duarte, Marco F., et al. "Single-pixel imaging via compressive sampling." IEEE signal processing magazine 25.2 (2008): 83-91.
>
> [4]Robucci, Ryan, et al. "Compressive sensing on a CMOS separable-transform image sensor." Proceedings of the IEEE 98.6 (2010): 1089-1101.
>
> [5]Mousavi, Ali, Gautam Dasarathy, and Richard G. Baraniuk. "Deepcodec: Adaptive sensing and recovery via deep convolutional neural networks." arXiv preprint arXiv:1707.03386 (2017).
>
> [6]Mousavi, Ali, Gautam Dasarathy, and Richard G. Baraniuk. "A data-driven and distributed approach to sparse signal representation and recovery." International Conference on Learning Representations. 2018.
>
> [7]Convolutional Neural Networks for Noniterative Reconstruction of Compressively Sensed Images
>
> [8]Li, Chengbo, et al. "An efficient augmented Lagrangian method with applications to total variation minimization." Computational Optimization and Applications 56.3 (2013): 507-530.
>
> [9]Dong, Weisheng, et al. "Compressive sensing via nonlocal low-rank regularization." IEEE Transactions on Image Processing 23.8 (2014): 3618-3632.
>
> [10]Goodfellow, Ian, et al. Deep learning. Vol. 1. No. 2. Cambridge: MIT press, 2016.
>
> [11]Kulkarni, Kuldeep, et al. "Reconnet: Non-iterative reconstruction of images from compressively sensed measurements." Proceedings of the IEEE Conference on Computer Vision and Pattern Recognition. 2016.

---

### Official Review · AnonReviewer1 · 2020-10-29
**Nonuniform Sampling Design for Compressive Sensing based on Gradient Descent of a Relaxation**

**Rating:** 5
**Confidence:** 4

**Review:**

The paper proposes a nonuniform sampling design scheme chosen using training data to reduce the computation of compressed sensing acquisition. The use of learning methods such as back propagation for the nonuniform sampling design problem is interesting. However, compressive sensing approaches in practice do not perform computation to obtain the measurement vector; instead, the sensors rely on custom hardware (such as the single pixel camera or the modulated wideband converter) to have the hardware act on the discretized signal according to the matrix design. Thus, considerations of the "sensing complexity" are moot in those cases, and the approach provided in this paper is only relevant in cases where custom sampling schemes can be designed (e.g., when the referred imaging sensors are used).

In these terms, the paper can be understood as the joint formulation of a nonuniform sampling and deep learning-based recovery of the sensed image. The authors provide numerical comparisons with random sampling and uniform subsampling, as well as with CS with random matrices. The use of uniform subsampling makes their approach comparable to existing approaches for image super-resolution based on deep learning, but this connection is not discussed.

There are some other minor issues with the paper:

* Claiming a complexity of O(1) for selective sensing ignores the fact that the number of measurements required for recovery is dependent on the signal length N, which is accounted for in the other complexities cited.
* Like selective sensing, sparse coding also does not require any prior knowledge of the sparsity structure of the data.
* Fourier (k-space) measurements can be obtained directly with optical hardware.
* The function o(.) has different inputs in different instances; please check for consistency.
* The relaxed formulation of the integer program does not specify the fact that the solution should be a subset of {0,...,M-1}; some of the entries of the set I might point to the same integer.
* The description of fine-tuning should be more detailed - is it simply more iterations of GD over Theta?
* The sentence "We also train the reconstruction networks in RS+Net from scratch based on the pre-trained selection indices in SS+Net" is not clear. Wouldn't this make the random samples not random?

---

> ### Author Response · Authors · 2020-11-24
> **Response to the comments(part1)**
>
> We thank the reviewer for reviewing the paper and providing constructive feedback.
>
> Reviewer’s comment: However, compressive sensing approaches in practice do not perform computation to ... where custom sampling schemes can be designed (e.g., when the referred imaging sensors are used).
>
> Response: We agree that computation complexity is not a good performance metric when custom hardware is used to implement the compressive sensing process in the analog domain. We respectfully argue that such custom hardware implementation inevitably increases the cost of the sensor. In addition, such custom hardware implementation is often specific to the sensor design, thereby cannot be generally applied to other sensors or applications. Differently, the selective sensing method is developed as a much more general low-cost sensing method mostly for resource-constrained, low-cost sensors. In this paper, we use image selective sensing just to provide the first proof-of-concept. Due to the computation-free nature of selective sensing, our method not only can provide a significant saving on the computational cost for those applications that must implement compressive/selective sensing in the digital domain (such as IoT sensors for time series) but also can reduce the cost of the custom hardware implementation required for those applications that implement the sensing in the analog domain, (such as the single-pixel camera).  Because the selective sensing operator (nonuniform subsampling) can be simply implemented as a static mask in the analog domain.  We revised the discussion on the necessity of selective sensing in section 1, paragraph 1 of our revised paper, to clarify this point to all the readers.
>
> Reviewer's comment: The use of uniform subsampling makes their approach comparable to existing approaches for image super-resolution based on deep learning, but this connection is not discussed.
>
> Response: We thank the reviewer for pointing out the topic of image super-resolution. The image super-resolution task is fundamentally different from the image reconstruction task of selective sensing in the following two aspects. First, images in super-resolution tasks are uniformly subsampled in the training phase, while images in selective sensing are nonuniformly subsampled. Therefore, the existing network structures for image super-resolution cannot be directly applied to perform the image reconstruction task in selective sensing. Second, the downsizing factor of images in super-resolution tasks is only up to 4x to the best of our knowledge in the existing literature. Differently, in selective sensing and reconstruction tasks, the nonuniformly subsampling factor (compression ratio) of images can have a much larger range (4-32x in our paper). We add the discussion on the differences between image super-resolution and selective sensing in section 2.4 of our revised paper to clarify this point to all readers.

---

> ### Author Response · Authors · 2020-11-24
> **Response to the comments(part 2)**
>
> Reviewer's comment:
>
> 1.Claiming a complexity of O(1) for selective sensing ignores the fact that the number of measurements required for recovery is dependent on the signal length N, which is accounted for in the other complexities cited.
>
> Response: We agree that the number of measurements required for recovery is dependent on the signal length N. But we respectfully argue that considering the signal length of N, the computational complexity of selective sensing is still O(1). This is simply because no data transformation nor data computation is involved in the sensing stage at all regardless of the signal length N. Selective sensing is performed as selecting values at specific positions based on a learned static index I. Such a process requires no computation upon the sensor data at all。  We are willing to further discuss with the reviewer regarding this claim and adjust the claim if it is commonly controversial. For now, we believe the claim is clear and justified.
>
> 2.Like selective sensing, sparse coding also does not require any prior knowledge of the sparsity structure of the data.
>
> Response: We agree. In section 2.1, paragraph 1 of our revised paper, we explicitly state that “sparse sensing requires a hand-crafted sparsity model of a signal as prior knowledge.”
>
> 3.Fourier (k-space) measurements can be obtained directly with optical hardware.
>
> Response: Thanks for pointing this out. Unfortunately, the complexity of the custom hardware[1] for implementing Fourier transformation is a deal-breaker for resource-constrained sensor devices and/or high-data-rate sensor devices dealing with high-dimensional signals. Differently, selective sensing works directly in the spatial domain, and the selective sensing operators require no computation upon the sensor data at all. We add the discussion on custom hardware for implementing Fourier transformation in section 2.1 of our revised paper to clarify this point to all readers.
>
> 4. The function $o(.)$ has different inputs in different instances; please check for consistency.
>
> Response: Thank you for catching this. We correct the inconsistency of $o$ in our revised paper.
>
> 5. The relaxed formulation of the integer program does not specify the fact that the solution should be a subset of ${0,...,M-1}$; some of the entries of the set I might point to the same integer.
>
> Response: The index vector $I$ is indeed not constrained to have all the entries to be unique in the training process. However, as we discovered after the training process, no index overlaps at any specific location(Visualization of the trained indices is shown in Figure 6,7,8 in the appendix). We believe that it is because the cases of overlapping indices are sub-optimal, and our algorithm already eliminated them in the training process.
>
> 6.The description of fine-tuning should be more detailed - is it simply more iterations of GD over Theta?
>
> Response: Yes, as we described in Algorithm 1, our algorithm simply optimizes $\Theta$ while keeping quantized $I$ to be fixed in the fine-tuning step.
>
> 7. The sentence "We also train the reconstruction networks in RS+Net from scratch based on the pre-trained selection indices in SS+Net" is not clear. Wouldn't this make the random samples not random?
>
> Response: This is an unfortunate misunderstanding. By using pre-trained selection indices from SS-Net as the sampling indices to train a reconstruction network from scratch(the sampling indices in fixed in the training process), we can reveal the reconstruction accuracy improvement contributed solely by the selection indices from selective sensing. We revise this sentence as“we also train two reconstruction networks from scratch at measurement rates of 0.03125 and 0.0125 with corresponding pre-trained selection indices from SS+Net remaining fixed in the training process, respectively.” in our revised paper to clarify this point.
>
> [1]Zhao, Wenfeng, et al. "On-chip neural data compression based on compressed sensing with sparse sensing matrices." IEEE transactions on biomedical circuits and systems 12.1 (2018): 242-254.
>
> [2]Hong, Tao, et al. "Optimized structured sparse sensing matrices for compressive sensing." Signal Processing 159 (2019): 119-129.
>
> [3]Duarte, Marco F., et al. "Single-pixel imaging via compressive sampling." IEEE signal processing magazine 25.2 (2008): 83-91.
>
> [4]Robucci, Ryan, et al. "Compressive sensing on a CMOS separable-transform image sensor." Proceedings of the IEEE 98.6 (2010): 1089-1101.

---

### Official Review · AnonReviewer4 · 2020-10-29
**Lack of novelty hinders the acceptance**

**Rating:** 4
**Confidence:** 5

**Review:**

This paper proposes to jointly optimize non-uniform subsampling pattern as well as the reconstruction network to perform image compressed sensing. It shows using learned selective sensing can significantly improve the reconstruction accuracy over the (random Gaussian) compressed sensing and uniform subsampling counterparts.

However, the novelty of the proposed method is very limited. The idea of using continuous interpolation on the discrete indices/grid, i.e. the main technical contribution in this paper, has already been proposed for CS-MRI sensing pattern design [1]. Although it is conducted on the k-space, applying this technique on image pixel space is straightforward and trivial. Besides, the comparison to a closely related work [2] is missing -- the probabilistic subsampling could also be used for the problem this paper tackles.
Finally, since the selective sensing pattern is learned from data, it's more fair to compare it with the learned compressed sensing counterpart, as exploited in the journal version of the ReconNet [3] used in this paper.

[1] PILOT: Physics-Informed Learned Optimized Trajectories for Accelerated MRI
[2] Deep probabilistic subsampling for task-adaptive compressed sensing
[3] Convolutional Neural Networks for Noniterative Reconstruction of Compressively Sensed Images

---

> ### Author Response · Authors · 2020-11-24
> **Response to the comments**
>
> We thank the reviewer for reviewing the paper and providing constructive feedback.
>
> Reviewer’s comment: However, the novelty of the proposed method ... is straightforward and trivial.
>
> Response: We respectfully disagree with this as the method in [1] and selective sensing are developed for different application scenarios with very distinct constraints on the computational cost of sensing. The method in [1] is for MRI imaging, while selective sensing is for resource-constrained IoT applications. As stated in [1], the technique proposed in [1] relies on non-uniform fast Fourier transformation (FFT) and the non-uniform inverse FFT techniques proposed in [4]. As we explained in the response to reviewer 3, the technique proposed in [1] that deals with much sparser signals in the frequency domain is therefore insufficient for dealing with dense signals directly in the spatial domain.  Additionally, due to the high computational cost of non-uniform FFT and nonuniform inverse FFT ($O(Nlog(N)+ Nlog(I/e)$), where $e$ is the precision of computations and $N$ is the number of nodes, the technique in [1] can hardly be affordable for resource-constrained sensor devices and/or high-data-rate sensor devices dealing with high-dimensional signals.
>
> Reviewer’s comment: Besides, the comparison to a closely related work [2] is missing -- the probabilistic subsampling could also be used for the problem this paper tackles.
>
> Response: There is a significant difference between DPS[2] and selective sensing. The nonuniform subsampling indices used by DPS are dynamically generated from a random distribution in the sensing process. In other words, the subsampling indices with respect to each signal is regenerated in practice. On the contrary, after the training process of selective sensing, the nonuniform subsampling indices are fixed. In other words, the subsampling indices with respect to each signal is static. The computation involved in dynamically generating the subsampling masks of DPS is unaffordable for resource-constrained sensor devices and/or high-data-rate sensor devices dealing with high-dimensional signals. Therefore, DPS can not be used to tackle the problem selective sensing is solving in this paper.  We add the discussion on the difference between DPS and selective sensing in section 2.1 in our revised paper.
>
> Reviewer's comment: Finally, since the selective sensing pattern is learned ... used in this paper.
>
> Response: We appreciate the related work [3] pointed out by the reviewer. However, we respectfully argue that the comparison between selective sensing and compressive sensing with random Gaussian matrices is fairer from the following three aspects. 1. The goal of our comparison study is to reveal the performance difference contributed by the differences in the sensing scheme rather than the reconstruction networks. So it is important to compare different sensing frameworks with the same reconstruction network settings. The co-trained algorithm in [3] is specific to the network structure in [3]. To the best of our knowledge, there is no general co-training algorithm that can be applied to various data-driven compressive sensing reconstruction networks. 2. Random Gaussian matrices are the most widely used sensing matrices in the domain of data-drive compressive sensing on images regardless of specific subsequent reconstruction network structures. Therefore, it is the most representative sensing matrices in this domain. 3. We conduct the experiments to compare the reconstruction accuracy of co-trained ReconNet[3] and the original ReconNet[4](with a random Gaussian sensing matrix) on CIFAR10 datasets at measurement rates of 0.125 and 0.25 to further consolidate our first point. The average improvements of [3] over [4] in PSNR/SSIM are 0.06 dB/0.007 and 0.83 dB/-0.0004, respectively. On the contrary, the improvement of selective sensing over compressive sensing with original ReconNet[4] 6.08/0.038 and 1.55/0.003, respectively. Compared to the reconstruction PSNR improvement contributed by the selective sensing operator, we consider the improvement of reconstruction PSNR/SSIM contributed by co-training as a minor improvement in the case of ReconNet. We add the discussion on the co-training algorithms proposed in the domain of data-driven compressive sensing in section 2.3 of our revised paper.
>
> [1] PILOT: Physics-Informed Learned Optimized Trajectories for Accelerated MRI
>
> [2] Deep probabilistic subsampling for task-adaptive compressed sensing
>
> [3] Convolutional Neural Networks for Noniterative Reconstruction of Compressively Sensed Images
>
> [4] Kulkarni, Kuldeep, et al. "Reconnet: Non-iterative reconstruction of images from compressively sensed measurements." Proceedings of the IEEE Conference on Computer Vision and Pattern Recognition. 2016.
>
> [5] Fast Fourier transforms for nonequispaced data

---

### Official Review · AnonReviewer3 · 2020-10-31
**An interesting approach to sampling point selection, but lacks context**

**Rating:** 4
**Confidence:** 5

**Review:**

The paper provides a data-driven approach to the selection of sampling points for compressive sensing. This is not a new problem, and has been studied for a while now. There is a large literature on it. For example, see below, references within and papers citing those:
- L. Baldassarre, Y. Li, J. Scarlett, B. Gözcü, I. Bogunovic and V. Cevher, "Learning-Based Compressive Subsampling," in IEEE Journal of Selected Topics in Signal Processing, vol. 10, no. 4, pp. 809-822, June 2016, doi: 10.1109/JSTSP.2016.2548442.
- Gözcü, Baran, et al. "Learning-based compressive MRI." IEEE transactions on medical imaging 37.6 (2018): 1394-1406.
- Bahadir, Cagla Deniz, Adrian V. Dalca, and Mert R. Sabuncu. "Learning-based optimization of the under-sampling pattern in MRI." International Conference on Information Processing in Medical Imaging. Springer, Cham, 2019.

The authors propose a relatively ad hoc relaxation of the problem to a continuous set instead of an index and then perform the selection there. However, there is no discussion on how this relaxed problem is in fact quite non-convex and how this choice of relaxation introduces a significant number of local minima. The comparison the authors offer is with very simplistic algorithms, rather than the more sophisticated ones in the literature.

Overall, this is an interesting approach that I think deserves to be eventually published. However, the paper needs significant work and comparison with the state of the art in the literature.

---

> ### Author Response · Authors · 2020-11-24
> **Response to the comments**
>
> We thank the reviewer for reviewing the paper and providing constructive feedback.
>
> Review comments: “The paper provides a data-driven approach to the selection of sampling points for compressive sensing.”
>
> Response:  We want to comment on that by highlighting the significant difference between selective sensing and compressive sensing. The nonuniform subsampling (selection) operation in selective sensing is computation-free, which is critical for resource-constrained sensor devices or high-data-rate sensor devices dealing with high-dimensional signals. We respectfully argue that the application of compressive sensing is limited in such scenarios from the following two aspects: 1. When the compressive sensing is performed in the digital domain as required in many IoT applications, the computational cost can be too high to be affordable for resource-constrained sensor devices, especially when the signal dimension $n$ is high and/or a data-driven sensing matrix (real-valued) is used; 2. When the compressive sensing is implemented in the analog domain (such as the single-pixel camera),  the custom hardware implementation inevitably increases the cost of the sensor and is often specific to the sensor design, thereby cannot be generally applied to other sensors or applications.
> We add a new discussion to Section 1, paragraph 1 of the revised paper to clarify this point to all readers.
>
> Reviewer’s comment: "This is not a new problem, and has been studied for a while now. There is a large literature on it..."
>
> Response: We appreciate the related references listed by the reviewer, and we add the discussion on the differences between selective sensing and them in section 2 of our revised paper. The major difference between selective sensing and [1,2,3] is the domain of sensing Selective sensing is performed in the original domain of signals directly. On the contrary, [1,2,3] all perform the sensing in a transformed domain (the transformation matrix is an orthonormal matrix in complex values). The computation needed for the transformation makes the computational cost of sensing in [1,2,3] significantly higher than selective sensing despite the details of actual hardware implementation. Additionally, as many spatial-domain signals are much sparser in the frequency domain, we respectfully argue that the existing nonuniform subsampling approaches performed in k-space are insufficient for dealing with dense signals directly in the spatial domain.
>
> Reviewer’s comment: "The authors propose a relatively ad hoc relaxation of the problem to a continuous set instead of an index and then perform the selection there. However, there is no discussion on how this relaxed problem is in fact quite non-convex and how this choice of relaxation introduces a significant number of local minima. The comparison the authors offer is with very simplistic algorithms, rather than the more sophisticated ones in the literature."
>
> Response: We appreciate the missing discussion pointed out by the reviewer, and we agree that the optimization problem (2) in the paper is essentially nonconvex, and the solution can be just local minima. The gap between local minima and global minima of neural networks remains an open field of research[2]. Even so, our empirical results show that our algorithm can consistently find a group of index $I$ that has a significantly better sensing performance than random indices, uniform indices, and random Gaussian matrices in terms of the reconstruction accuracy under different circumstances. We add a new discussion to section 4.4 of our revised paper to clarify this point to all readers.
>
> Reviewer's comment: Overall, this is an interesting approach that I think deserves to be eventually published. However, the paper needs significant work and comparison with the state of the art in the literature.
>
> Response: Thank you for the positive remarks.
>
> [1] L. Baldassarre, Y. Li, J. Scarlett, B. Gözcü, I. Bogunovic and V. Cevher, "Learning-Based Compressive Subsampling," in IEEE Journal of Selected Topics in Signal Processing, vol. 10, no. 4, pp. 809-822, June 2016, doi: 10.1109/JSTSP.2016.2548442.
>
> [2] Gözcü, Baran, et al. "Learning-based compressive MRI." IEEE transactions on medical imaging 37.6 (2018): 1394-1406.
>
> [3] Bahadir, Cagla Deniz, Adrian V. Dalca, and Mert R. Sabuncu. "Learning-based optimization of the under-sampling pattern in MRI." International Conference on Information Processing in Medical Imaging. Springer, Cham, 2019.
>
> [4] Huijben, Iris AM, Bastiaan S. Veeling, and Ruud JG van Sloun. "Deep probabilistic subsampling for task-adaptive compressed sensing." International Conference on Learning Representations. 2019.
>
> [5] Goodfellow, Ian, et al. Deep learning. Vol. 1. No. 2. Cambridge: MIT press, 2016.

---

### Public Comment · ~Iris_A.M._Huijben1 · 2020-11-13
**Related work of interest to you**

First of all, congrats with this submisson! It's nice to see how the research interest for learned (discrete) subsampling at the sensor side is expanding now.

Out of curiosity I've a question regarding your sample selector I. Is it possible in the current framework that two learned indices in I are so close that after rounding they are rounded to the same integer index, resulting in the same sample/pixel to be sampled twice or even multiple times?
If not, how do you prevent this or is the model just teaching itself that this is suboptimal to do?

Also you might be interested in this work where we performed learned subsampled at the sensor side for radio-frequency ultrasound data, combined with joint learning of a downstream task model:
"Learning Sub-Sampling and Signal Recovery with Applications in Ultrasound Imaging"
https://ieeexplore.ieee.org/document/9138467

Good luck with the rebuttal phase and let's hope to meet in person in a conference at some moment when the Covid situation is all over!

---

> ### Author Response · Authors · 2020-11-24
> **Answer to the question**
>
> Thanks for the supportive comment. This is an interesting question. The index vector $I$ is indeed not constrained to have all entries to be unique in the training process. However, as we discovered after the training process, no index overlaps at any specific location. We believe that it is because the cases of overlapping indexes are sub-optimal, and our algorithm already eliminated them in the training process.

---

### Decision · Program_Chairs · 2021-01-07
**Final Decision**

**Decision:**

Reject

**Comment:**

The reviewers all agree that the problems studied in this paper are interesting, and the solutions provided are reasonable.  However qualitative and quantitative comparisons to state of the art methods are missing, and the sensing model assumed by the paper needs to be more well motivated.